# The Effects of Grit, Emergency Nursing Competency, and Positive Nursing Organisational Culture on Burnout Among Nurses in the Emergency Department

**DOI:** 10.3390/bs15040486

**Published:** 2025-04-08

**Authors:** Su-Young Moon, Hyung-Ran Park

**Affiliations:** College of Nursing, Research Institute of Nursing Science, Chungbuk National University, Cheongju 28644, Chungbuk, Republic of Korea; tndudsy@naver.com

**Keywords:** emergency, nurses, grit, emergency nursing competency, positive nursing organisational culture, burnout

## Abstract

This descriptive cross-sectional study investigated the relationship between grit, emergency nursing competency, positive nursing organisational culture, and burnout among emergency department nurses and identified the factors influencing burnout. The study participants were 176 nurses from one tertiary and six general hospitals in Chungcheong-do, South Korea. Data were collected from 18 July to 10 August 2024 and analysed using hierarchical multiple regression. Burnout was negatively correlated with grit (*r* = −0.57, *p* < 0.001), emergency nursing competency (*r* = −0.41, *p* < 0.001), and positive nursing organisational culture (*r* = −0.60, *p* < 0.001). It was also negatively correlated with the subdomains of positive nursing organisational culture: active leadership of nurse managers (*r* = −0.46, *p* < 0.001), pursuit of shared values (*r* = −0.55, *p* < 0.001), trust-based organisational relationship formation (*r* = −0.62, *p* < 0.001), and fair management systems (*r* = −0.55, *p* < 0.001). The regression analysis showed two variables that significantly affected burnout—trust-based organisational relationship formation in positive nursing organisational culture (β = −0.31, *p* = 0.012) and grit (β = −0.29, *p* < 0.001); their explanatory power was 44.0% (F = 18.19, *p* < 0.001). Thus, positive nursing organisational culture and grit were major factors affecting burnout. Therefore, to prevent and effectively manage burnout among emergency department nurses, nursing organisations must create a positive organisational atmosphere based on mutual respect and trust, promoting active participation in work, decision-making, and collaboration.

## 1. Introduction

The emergency department (ED) is characterised by overcrowding due to an endless stream of patients ([39]). In this context of overcrowding, ED nurses often encounter challenges such as time constraints, increased responsibilities, and longer shift times; they may thus experience burnout and frustration at work ([24]). In particular, because patients visiting the ED often experience severe acute diseases or life-threatening trauma, their condition can deteriorate rapidly ([21]). These changes expose healthcare providers to the risk of burnout daily ([21]). The proportion of ED healthcare providers who have experienced burnout is 43%, which is the highest among nurses, at 47% ([2]). Compared to burnout rates of 37.4% for healthcare providers during the COVID-19 pandemic ([7]), and 30% for nurses in the last 10 years ([20]), this indicates a high burnout rate in the ED.

Since May 2019, burnout has been included in the World Health Organisation’s 11th International Classification of Diseases (ICD-11) as an occupational phenomenon ([45]). Burnout is a syndrome characterised by emotional exhaustion, depersonalisation, and reduced personal accomplishment ([38]). Emotional exhaustion, the first component, is a state of feeling low energy, despair, and tension due to excessive work-related demands. Depersonalisation, the second component, is a state marked by a numb, passive attitude when dealing with colleagues and other people in one’s environment. Reduced personal accomplishment, the third component, is a state of persistently low confidence and a feeling of incapability with regard to work ([38]). This syndrome can have various effects on ED nurses. Nurses who experience burnout can have poor health outcomes, including physical frailty, insomnia, hostility, hypersensitivity, depression, and drug abuse ([35]). This condition can not only lead to risks such as medical errors or infections among patients but also reduce patient safety and the quality of care ([35]). Ultimately, healthcare organisations face increased absence from work and lower work productivity ([21]). As such, a better understanding of burnout is needed to alleviate these effects.

According to theoretical discussions around burnout in nursing, burnout is associated with high work burdens, low staff numbers, long work times, low autonomy and sense of control, time pressures, high emotional burdens, role conflicts, negative nurse–physician relationships, and poor organisational support ([16]). Burnout is more likely caused by insufficient job resources than by high job demands. Job resources at the personal level include technical knowledge and skills and positive psychological capital (self-efficacy, resilience, and optimism), and at the organisational level, they include temporal flexibility, a safe environment, support from managers and colleagues, sufficient resources, autonomy, and compensation. These resources can help prevent burnout ([19]). Previous studies have also shown that burnout is associated with the personal resources of grit ([4]; [8]; [22]), nursing competence ([5]; [27]; [42]; [46]), and the organisational resource of organisational culture ([4]).

Grit, a property of one’s psychological character, can alleviate burnout by boosting one’s passion and persistence toward accomplishment, even when faced with experiences of despair and hardship ([18]). A study of physicians noted that burnout could be reduced by remaining persistent in reaching one’s goals when exposed to challenging situations ([8]; [22]). Studies on grit and academic burnout in nursing and medical students have shown that grit plays a positive role in personal psychological health and is a reliable index that serves as a protective factor against burnout ([28]; [30]). Even though there have been various studies on the relationship between grit and burnout focusing on students and physicians, there have not been any studies on nurses or ED nurses in particular.

According to a meta-analysis focused on burnout in ED nurses, insufficient competency in ED nursing work increases emotional exhaustion and induces burnout ([21]). One study of novice nurses in China showed that those who could not adequately provide care due to low competency experienced high burnout ([46]). In another study investigating factors affecting burnout in nurses from 22 organisations in Indonesia, competence showed a negative correlation with burnout ([27]). In nurses caring for patients needing terminal care as well, psychological burnout was found to be negatively correlated with end-of-life care competency ([42]). Thus, nurses can reduce burnout by demonstrating competency—the ability to provide quality care to patients in various situations ([27]). In systematic literature reviews, nursing competency has been defined as the ability to integrate and apply the knowledge, skills, and attitudes needed to provide patients with effective care ([5]; [50]). Nursing competency in the ED is the ability of nurses, as a staff team, to identify a patient’s condition promptly, perform triage, and provide acute and critical care ([44]). In particular, emergency nursing competency constitutes the ability to consistently practice clinical professionalism, effective communication, and teamwork and tap into the resources, environment, and legal values in an emergency setting ([44]). This competency improves work performance and can help reduce burnout.

Regarding organisational resources that affect burnout, a comprehensive analysis revealed that organisational culture in nursing had important effects on nurses’ turnover, leadership, management, well-being, work environment challenges, and healthcare delivery ([6]). The nursing organisational culture includes experiences, values, beliefs, rules, policies, missions, and ethics formed and shared within the nursing organisation, management and leadership of the organisation, and the organisational structure ([1]). This serves as both a guideline that determines how the nursing organisation will function and how individuals will behave, as well as core research that affects the organisation’s achievements and individuals’ adaptability ([32]). In the relationship between organisational culture and burnout, a negative organisational culture can induce burnout due to role conflict and low support. However, a positive organisational culture can increase autonomy and prevent burnout via an environment that supports and empowers its members ([32]).

A positive nursing organisational culture is characterised by clear communication, nurses’ participation in major decision-making processes alongside leadership, strong support from the organisation and managers, a transparent work system, plentiful opportunities for professional development, ample opportunities for continuing education for self-growth, strong cooperation and teamwork, interest in the personalised needs of nurses, values and respect for professionalism, and adequate staffing and resources ([32]). These factors can help alleviate work-related stress in nurses. Among previous studies, one study of workers at a U.S. healthcare organisation showed that those with a more positive perception of the organisational culture reported lower burnout ([26]). In a study of nurses in Iran, organisational culture was negatively correlated with burnout ([1]). A study of nurses in Turkey also showed a negative correlation between organisational culture and burnout ([41]). In one study of ED nurses, organisational culture factors such as teamwork, support from nurse managers, compensation, and other job resources showed significant effects on burnout ([39]).

Being provided ample opportunities to make intentional efforts toward a goal allows one to develop grit, and this strategy could help alleviate burnout ([34]) in ED nurses. Although grit is a personal characteristic, it can be shaped through education and experience ([29]). Furthermore, as organisational culture is a factor that changes over time ([1]), it is vital to conduct research to understand the relationship between burnout and positive nursing organisational culture in recent ED nursing organisations, which have been affected by various infectious diseases. Therefore, to alleviate or prevent burnout in ED nurses, it is crucial to investigate the relationship between grit, emergency nursing competency, and positive organisational culture.

### Objectives

The aim of this study was to investigate the influence of ED nurses’ grit, emergency nursing competency, and positive nursing organisational culture on their burnout experiences. The research questions were as follows: (1) What are the levels of burnout experienced by ED nurses and the related differences according to their general characteristics? (2) What are the levels of grit, emergency nursing competency, and positive nursing organisational culture? (3) Is there a relationship between grit, emergency nursing competency, positive nursing organisational culture, and burnout? (4) What are the factors influencing burnout in ED nurses?

## 2. Methods

### 2.1. Study Design

This descriptive cross-sectional study aimed to identify the factors influencing burnout among ED nurses.

### 2.2. Participants and Setting

The participants in this study were nurses working for at least six months in the ED of one tertiary or one of six general hospitals in Chungcheong-do, South Korea. The required number of participants was calculated to be 166 nurses using G*power 3.1.9.7 for multiple regression with an explanatory power of 0.90, a significance level of 0.05, [14]’s ([14]) medium effect size of 0.15, and 14 predictors (eight general characteristics, two independent variables, and one independent variable consisting of four subfactors). To account for a 10% dropout rate, we distributed questionnaires to 185 nurses, and we received 184 responses (99%). After excluding eight insincere responses, we included the remaining 176 responses in the final analysis.

### 2.3. Variables and Measurement

#### 2.3.1. General Characteristics

General characteristics consisted of sex, age, marital status, education level, type of emergency organisation, clinical experience, work experience in the ED, and monthly income.

#### 2.3.2. Grit

Grit was measured using the Clinical Nurses Grit Scale (CN-GRIT), developed and validated by [40] ([40]), after obtaining approval from the authors. This instrument consists of 14 items, including questions covering aspects such as persistence to achieve long-term goals, passion for becoming a nursing professional, and patient-oriented intrinsic motivation. Each item is rated on a 4-point Likert scale, with higher scores indicating higher grit. In terms of reliability, Cronbach’s α was 0.91 at the time of development ([40]) and 0.90 in our study.

#### 2.3.3. Emergency Nursing Competency

We measured the ED nurses’ nursing competency and behaviour index, developed by [49] ([49]), using an instrument revised by [12] ([12]) for ED nurses. This instrument consists of 33 items, including aspects related to clinical judgement and coping, ability to handle ward work, flexibility, resource management, self-confidence, cooperation, professional development ability, patient orientation, ethical values orientation, influence, developing others, and self-control. Each item is rated on a 5-point Likert scale, with higher scores indicating better emergency nursing competency. In terms of reliability, Cronbach’s α was 0.96 in the study by [12] ([12]) and 0.96 in our study.

#### 2.3.4. Positive Nursing Organisational Culture

We measured positive nursing organisational culture using the positive nursing organisation culture measurement tool developed and validated by [31] ([31]), after obtaining approval from the authors. This instrument consists of 26 items; it is divided into four subdomains: positive leadership of the nursing unit manager, pursuit of common values, forming organisational relationships based on trust, and a fair management system. Each item is rated on a 5-point scale, with higher scores indicating a more positive perception of nursing organisational culture. In terms of reliability, at the time of development ([31]), Cronbach’s α was 0.95 for the whole scale and 0.95, 0.88, 0.92, and 0.83 for each subfactor. In our study, Cronbach’s α was 0.97 for the whole scale and 0.96, 0.90, 0.92, and 0.88 for each subfactor.

#### 2.3.5. Burnout

To measure burnout, we purchased and used the Maslach Burnout Inventory (MBI), developed by [37] ([37]), adapted into Korean for social workers by [25] ([25]), and revised by [47] ([47]) for use with ED nurses. This instrument consists of 26 items, including the subscales of emotional exhaustion, depersonalisation, and reduced personal accomplishment. Each item is rated on a 5-point Likert scale, with higher scores indicating a higher level of burnout. In terms of reliability, Cronbach’s α was 0.90 in a study of ED nurses by [47] ([47]) and 0.92 in our study.

### 2.4. Data Collection

After receiving approval from the institutional review board, we collected data between 18th July and 10th August 2024. We visited the study hospitals directly to ask for cooperation with data collection from the heads of the nursing department and the ED. We visited the ED during nurses’ shift times. A written explanation sheet was provided to explain the study objectives and methods and obtain the participants’ written informed consent to participate in the study. Data were collected using self-report questionnaires. To protect the participants’ privacy, the filled questionnaires were returned in a sealed envelope and marked with an identification number. The participants were provided with a small gift as a token of gratitude for their participation.

### 2.5. Analysis

The collected data were analysed using descriptive statistics, independent t-tests, and one-way ANOVA, and the software used was SPSS Statistics 29.0. The correlations of burnout with grit, emergency nursing competency, and positive nursing organisational culture were analysed using Pearson’s correlation. To identify the factors affecting burnout, we performed hierarchical multiple regression analysis to control for the general characteristics of the participants from the seven study hospitals. The normal distribution of burnout based on the general characteristics of the participants was analysed by conducting the Shapiro–Wilk test. The normal distribution was confirmed at *p* > 0.05.

### 2.6. Ethical Considerations

This study was conducted after review and approval by the institutional review board at the authors’ affiliated institution (CBNU-2024-A-020). The participants were provided with a written explanation of the study objectives and methods, and they voluntarily provided their informed consent by signing two copies of the consent form. One copy was kept by the investigators, and one copy was provided to the individual participants. The explanation sheet described the study objectives and methods, as well as ethical considerations such as the protection of personal information, anonymity, privacy, and the risks and benefits of participating in the study. It explained that the participants could withdraw from the study at any time without suffering any disadvantages and that the collected data would be used and published only for research purposes. The collected questionnaires were assigned an arbitrary subject identification number and kept in a locked filing cabinet. The questionnaires were also digitalised, and they were stored and maintained on a separate mobile device with password protection.

## 3. Results

### 3.1. Differences in Burnout According to the Participants’ General Characteristics

The average age of the participants was 30.13 ± 6.07 years, with an average clinical experience of 6.29 ± 5.74 years and ED experience of 4.71 ± 4.24 years. Among the total 176 participants, 125 (71.0%) were female and 51 (29.0%) were male. Burnout was significantly higher in the female participants (*t* = 2.33, *p* = 0.021). Regarding marital status, 134 (76.1%) were unmarried and 42 (23.9%) were married. Burnout was significantly higher in the unmarried group (*t* = 3.05, *p* = 0.003) (Table 1).

### 3.2. Levels of Grit, Emergency Nursing Competency, and Positive Nursing Organisational Culture

The item mean grit score was 3.03 ± 0.43 (range 1.00–4.00), and the item mean emergency nursing competency score was 3.63 ± 0.59 (range 1.82–5.00). The item mean positive nursing organisational culture score was 3.88 ± 0.67 (range 1.04–5.00), with the mean scores for its subdomains as follows: forming organisational relationships based on trust (3.94 ± 0.68), pursuit of common values (3.91 ± 0.69), positive leadership of the nursing unit manager (3.88 ± 0.85), and a fair management system (3.67 ± 0.80). The item mean burnout score among the emergency room nurses was 2.56 ± 0.60 (range 1.23–4.14), with the following mean scores for its subscales: emotional exhaustion (2.76 ± 0.60), depersonalisation (2.31 ± 0.78), and personal accomplishment (2.50 ± 0.60; Table 2).

### 3.3. Correlations Among Grit, Emergency Nursing Competency, and Positive Nursing Organisational Culture

Burnout was negatively correlated with grit (*r* = −0.57, *p* < 0.001), emergency nursing competency (*r* = −0.41, *p* < 0.001), and positive nursing organisational culture (*r* = −0.60, *p* < 0.001). Additionally, burnout was negatively correlated with the subdomains of positive nursing organisational culture, including positive leadership of the nursing unit manager (*r* = −0.46, *p* < 0.001), pursuit of common values (*r* = −0.55, *p* < 0.001), forming organisational relationships based on trust (*r* = −0.62, *p* < 0.001), and a fair management system (*r* = −0.55, *p* < 0.001) (Table 3).

### 3.4. Factors Affecting the Levels of Burnout

Hierarchical multiple regression analysis was conducted to identify the factors influencing burnout among emergency room nurses (Table 4). Grit, emergency nursing competency, and the subfactors of positive nursing organisational culture—positive leadership of the nursing unit manager, pursuit of common values, forming organisational relationships based on trust, and fair management system—were entered as continuous independent variables. Categorical variables, including gender and marital status, which were associated with significant differences in burnout experiences, were dummy coded. Tolerance was above the cutoff of 0.1 (range 0.22–0.90), the variance inflation factor (VIF) was below the cutoff of 10 (range 1.11–4.52), and the Durbin–Watson coefficient was close to 2 (value 2.191). These values confirmed the absence of multicollinearity between the independent variables.

In Model 1, the participants’ general characteristics were included as control variables. These variables explained 7.1% of the variance in burnout, and gender (female; β = 0.18, *p* = 0.017) and marital status (unmarried; β = 0.23, *p* = 0.002) had significant effects on burnout. In Model 2, when the personal factors of grit and emergency nursing competency were added, the variables explained 34.1% of the variance in burnout (F = 23.64, *p* < 0.001), and grit was found to have a significant effect on burnout (β = −0.48, *p* < 0.001). In Model 3, the subfactors of positive nursing organisational culture were added hierarchically as organisational factors. Here, the factors with significant effects on burnout in the ED nurses were the positive nursing organisational culture subfactor of forming relationships based on trust (β = −0.31, *p* = 0.012) and grit (β = −0.29, *p* < 0.001). The variables in this model explained 44.0% of the variance in burnout in the ED nurses (F = 18.19, *p* < 0.001).

## 4. Discussion

In this study, we investigated the factors affecting burnout in ED nurses; we found that grit and forming relationships based on trust, as a subfactor of positive nursing organisational culture, had significant effects on burnout. These two variables explained 44.0% of the variance in burnout.

In our study, forming organisational relationships based on trust, a subfactor of positive nursing organisational culture, was the factor with the strongest effect on burnout. Trust-based relationship building is a matter of teamwork and pride, involving teamwork building between members of the nursing organisation based on trust, as well as empathy and consolation between colleagues during work-related conflicts, resulting in improved work satisfaction ([31]). These results align with several previous reports indicating that organisational teamwork is related to burnout. In a study on organisational culture and burnout among healthcare providers in the Netherlands, among subfactors of positive organisational culture, the relational atmosphere in the workplace showed a significant negative correlation with burnout ([33]). In another study of healthcare system employees in the United States, burnout was lower when there was a positive organisational culture that valued teamwork and collaboration, where colleagues’ emotions were considered when working together ([26]). In a study of nurses in Oman, teamwork and job satisfaction were found to affect burnout ([3]). Studies of ED nurses ([39]) and nurses in South Korea ([11]) have also identified teamwork as a major factor in reducing burnout. Thus, maintaining relationships with colleagues based on mutual respect and cooperation has been shown to reduce burnout during nursing work. In terms of strategies to improve teamwork in nursing organisations, enhancing the ability of the team and its members to cope flexibly with dynamic work environments, such as the ED, can help prevent burnout ([11]). Given that teamwork in nursing organisations is an important factor affecting burnout in nurses, effective interventions to alleviate burnout will require an organisational approach. One systematic literature review proposed that, when developing interventions for burnout, the integration of individual-level and organisational-level approaches should be considered ([48]). Individual-level approaches have been shown to have a rapid effect on burnout and are impactful in reducing burnout in the long term when combined with organisational-level approaches ([13]). In this context, various interventions have been proposed to improve the work environment and reduce stress to promote teamwork within organisations.

We also found that grit had a significant effect on burnout, consistent with previous studies. Burnout and grit were reported to be inversely correlated in a study of internal medicine residents ([8]) and in a study of orthopaedic residents and specialists ([22]). Studies on medical and nursing students have also shown that grit is an important personal resource related to burnout ([28]; [30]; [34]). These findings show that grit is a protective factor that can predict burnout, suggesting that the extent of burnout can be regulated by the unique personal characteristic of grit. Training to develop and improve grit provides the opportunity to not only temporarily cope with burnout but also actively prevent it ([28]). Therefore, to identify the early symptoms of burnout and prevent burnout in ED nurses, it is vital to implement programs to improve grit ([22]). In particular, although grit is a personal characteristic, to develop grit in individuals who share and challenge the organisation’s objectives, it is important for grit to be increased throughout the work environment. In other words, forming a culture within nursing organisations that enables nurses to achieve a high level of grit could be more effective for overcoming burnout than simply training individual nurses ([17]). Furthermore, individuals with higher resilience might be able to develop adaptive mechanisms; thus, resilience-based interventions from nursing students would help mitigate burnout ([29]).

Emergency nursing competency, however, was not found to have a significant effect on burnout in our study. This finding differs from previous studies that highlighted competency in nursing work as a negative predictive factor for burnout ([27]; [46]). For nurses, strong nursing competency can increase the work burden if it leads to more responsibilities, especially in complex situations ([2]). Further studies are needed to clarify the relationship between these two factors.

Although gender and marital status were significant factors in the univariate analysis in Model 1, they were not significant in the final multivariate analysis in Model 3 for burnout. This result is consistent with the findings of a previous systematic review focused on emergency department healthcare workers, which revealed that burnout experiences did not significantly differ based on gender ([2]). However, different results have been reported in studies involving clinical nurses in Korea ([10]), city hospital nurses in Turkey ([5]), and clinical nurses in China ([46]). Although the evidence on gender is inconsistent, this result is not entirely inexplicable; globally, female nurses are likely more vulnerable to and influenced by burnout due to diverse reasons such as societal roles and expectations, emotional factors, and family responsibilities, among others. Therefore, female nurses should be considered an important population in burnout management efforts. Additionally, the finding on marital status is consistent with the results of previous research; higher scores were reported in the unmarried participant group, but it was not a predictor of burnout in the univariate analysis ([10]), and there were no clear differences associated with burnout ([36]; [46]). Further research exploring the link between marital status and burnout is recommended.

Our study has important implications for nursing work and organisations. Strategic support will be required to form nursing organisational cultures that can improve teamwork between members. In addition, even though grit is a property of personal character involving persistence and effort, rather than an approach that simply targets individual nurses, it is important to develop a cultural environment within organisations conducive to improving grit. Additionally, we considered the intrinsic factors that might explain the varying levels of burnout reported in previous research ([9]; [15]; [23]; [43]). Accordingly, workplace factors should be considered along with intrinsic psychological factors.

The main strength of our study is that we identified positive nursing organisational culture as a factor affecting burnout in ED nurses, and we explored its effects on burnout. We believe that this study fills a gap, since, to the best of our knowledge, no prior studies have explored burnout in nurses or ED nurses. Another strength of our study is that we analysed the effects of both personal and organisational factors on burnout.

### Limitations

As our study only included ED nurses from a region in South Korea, there could be limitations in generalising our findings. In addition, we used a cross-sectional design, so it will be important to investigate the long-term effects of organisational culture in later studies. Furthermore, because we did not use random sampling to select the participants and provided a small gift as a token of appreciation for their engagement, we cannot exclude the possibility of sampling bias. Moreover, the nurse population in Korea generally consists of a large number of female nurses, which could again lead to sampling bias. Finally, owing to the use of self-report questionnaires to collect the data, the participants’ subjective views could have been reflected in the results.

## 5. Conclusions

The highlight of this study is that we identified significant factors that help reduce burnout in ED nurses in South Korea. Burnout in ED nurses is related to the formation of organisational relationships based on trust, which is a subfactor of positive nursing organisational culture. In addition, we identified grit as a protective factor preventing burnout. We propose that ED nurses be provided with opportunities to participate in supportive programs focused on developing grit and fostering the teamwork required to form positive relationships within nursing organisations.

## Figures and Tables

**Table 1 behavsci-15-00486-t001:** Differences in burnout according to the participants’ general characteristics (N = 176).

Variables	Categories	*n*	%	Burnout	
				M ± SD	*t*/F(*p*)
Gender	Female	125	71.0	2.63 ± 0.61	2.33 (0.021)
	Male	51	29.0	2.40 ± 0.56	
Age (years)	20–<30	94	53.4	2.55 ± 0.52	2.24 (0.109)
	30–<40	69	39.2	2.63 ± 0.68	
	40≤	13	7.4	2.25 ± 0.70	
Marital status	Unmarried	134	76.1	2.64 ± 0.57	3.05 (0.003)
	Married	42	23.9	2.31 ± 0.65	
Educational status	Diploma	24	13.6	2.68 ± 0.60	0.61 (0.543)
	Bachelor’s degree	125	71.0	2.55 ± 0.60	
	Above master’s degree	27	15.4	2.51 ± 0.65	
Type of EM centre	Regional EM centre	71	40.3	2.61 ± 0.55	0.52 (0.597)
	Local EM centre	77	43.8	2.55 ± 0.68	
	Local EM institution	28	15.9	2.47 ± 0.60	
Total clinical career (years)	<5	82	46.6	2.54 ± 0.56	0.10 (0.903)
	5–<10	58	33.0	2.59 ± 0.57	
	10≤	36	20.4	2.57 ± 0.75	
Career in emergency department (years)	<5	108	61.4	2.52 ± 0.55	0.56 (0.570)
	5–<10	48	27.3	2.61 ± 0.69	
	10≤	20	11.3	2.65 ± 0.66	
Monthly income (10,000 won)	<300	79	44.9	2.65 ± 0.56	1.42 (0.244)
	300–<400	76	43.2	2.49 ± 0.59	
	400≤	21	11.9	2.48 ± 0.78	

Notes. *n*: frequency; EM: emergency medical; M: mean; SD: standard deviation.

**Table 2 behavsci-15-00486-t002:** Levels of grit, emergency nursing competency, and positive nursing organisational culture (N = 176).

Variables	Number of Items	Item Mean			Scale Range
		M ± SD	Min	Max	
Grit	14	3.03 ± 0.43	1.00	4.00	1–4
Emergency nursing competency	33	3.63 ± 0.59	1.82	5.00	1–5
Positive nursing organisational culture	26	3.88 ± 0.67	1.04	5.00	1–5
Positive leadership of the nursing unit manager	7	3.88 ± 0.85	1.00	5.00	1–5
Pursuit of common values	7	3.91 ± 0.69	1.00	5.00	1–5
Forming organisational relationships based on trust	8	3.94 ± 0.68	1.13	5.00	1–5
Fair management system	4	3.67 ± 0.80	1.00	5.00	1–5
Burnout	22	2.56 ± 0.60	1.23	4.14	1–5
Emotional exhaustion	9	2.76 ± 0.84	1.11	5.00	1–5
Depersonalisation	5	2.31 ± 0.78	1.00	4.40	1–5
Personal accomplishment	8	2.50 ± 0.60	1.00	4.00	1–5

Notes. M: mean; SD: standard deviation; Min: minimum; Max: maximum.

**Table 3 behavsci-15-00486-t003:** Correlations among grit, emergency nursing competency, and positive nursing organisational culture (N = 176).

Variables	Grit	ENC	PNOC					Burnout			
			Total	1.	2.	3.	4.	Total	1.	2.	3.
	*r* (*p*)										
Grit	1										
ENC	0.59 *	1									
PNOC	0.59 *	0.47 *	1								
1. Leadership	0.48 *	0.39 *	0.87 *	1							
2. Values	0.56 *	0.43 *	0.92 *	0.71 *	1						
3. Relationship	0.60 *	0.44 *	0.92 *	0.68 *	0.85 *	1					
4. System	0.48 *	0.44 *	0.88 *	0.69 *	0.77 *	0.78 *	1				
Burnout	−0.57 *	−0.41 *	−0.60 *	−0.46 *	−0.55 *	−0.62 *	−0.55 *	1			
1. Emotional exhaustion	−0.49 *	−2.50 *	−0.52 *	−0.40 *	−0.48 *	−0.56 *	−0.43 *	0.90 *	1		
2. Depersonalisation	−0.41 *	−0.34 *	−0.49 *	−0.36 *	−0.48 *	−0.47 *	−0.50 *	0.77 *	0.58 *	1	
3. Personal accomplishment	−0.47 *	−0.48 *	−0.44 *	−0.35 *	−0.38 *	−0.44 *	−0.44 *	0.73 *	0.43 *	0.42 *	1

Notes. ENC: emergency nursing competency; PNOC: positive nursing organisational culture. * < 0.001.

**Table 4 behavsci-15-00486-t004:** Factors influencing burnout among the nurses (N = 176).

Variables	Model 1				Model 2				Model 3			
	B	SE	β	*t* (*p*)	B	SE	Β	*t* (*p*)	B	SE	β	*t* (*p*)
Constant	2.15	0.11		19.03 (<0.001)	4.76	0.33		14.53 (<0.001)	5.19	0.31		16.56 (<0.0001)
Gender ^†^	0.23	0.10	0.18	2.41 (0.017)	0.14	0.08	0.10	1.64 (0.102)	0.08	0.08	0.06	0.97 (0.332)
Marital status ^†^	0.32	0.10	0.23	3.11 (0.002)	0.15	0.09	0.11	1.66 (0.098)	0.10	0.09	0.07	1.21 (0.229)
Grit					−0.68	0.11	−0.48	−6.23 (<0.001)	−0.42	0.11	−0.29	−3.65 (<0.001)
ENC					−0.10	0.08	−0.09	−1.19 (0.234)	−0.03	0.08	−0.03	−0.34 (0.732)
PNOC												
Leadership									0.02	0.06	0.03	0.33 (0.741)
Values									0.01	0.10	0.01	0.08 (0.934)
Relationship									−0.27	0.11	−0.31	−2.55 (0.012)
System									−0.12	0.08	−0.16	−1.62 (0.107)
F (*p*)	7.68 (0.001)				23.64 (<0.001)				18.19 (<0.001)			
R^2^ (adjusted R^2^)	0.082 (0.071)				0.356 (0.341)				0.466 (0.440)			
R^2^ change					0.275				0.110			
Tolerance					0.61–0.97				0.22–0.90			
VIF					1.03–1.65				1.11–4.52			

Notes. B: unstandardised regression coefficient; SE: standard error; β: standardised regression coefficient; ENC: emergency nursing competency; PNOC: positive nursing organisational culture; VIF: variance inflation factor; ^†^ dummy variable: gender_female (ref. = male), marital status_unmarried (ref. = married).

## Data Availability

Data are contained within the article.

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
