# Peer review of "The Effects of Grit, Emergency Nursing Competency, and Positive Nursing Organisational Culture on Burnout Among Nurses in the Emergency Department"

_behavsci, 2025, doi:10.3390/bs15040486_

Round 1
Reviewer 1 Report
Comments and Suggestions for Authors
This study addresses an important topic with significant implications for emergency nursing. However, before publication, substantial revisions are needed.
- Line 11: I recommend explicitly stating that the study is a "descriptive cross-sectional study" in the abstract to improve clarity for readers.
- Line 71: The sentence "One meta-analysis demonstrated that grit is associated with conscientiousness [19]." appears disconnected from the surrounding context. It would be beneficial to clarify its relevance to the study’s objectives.
- Lines 78-83: This paragraph would be more appropriately placed at the end of the introduction. Additionally, I recommend elaborating on the research questions and hypotheses in this section to create a clearer logical flow. Currently, the transition to the study’s aims is too brief and general (lines 133-135). More precise research questions or hypotheses should be explicitly stated.
- Lines 184-190: The burnout questionnaire includes three subscales, but their detailed results are not presented later in the paper. Furthermore, it would be useful to indicate whether there are commonly used cut-off scores for the Maslach Burnout Inventory and how the studied population compares to these thresholds.
- Lines 195-200: The methodology mentions that data collection occurred during nurses’ shift times and that participants received a small gift. These factors could have influenced the results. I suggest discussing this potential bias explicitly in the limitations section.
- Line 203: Please specify the exact type of ANOVA used. It is likely one-way ANOVA, but confirmation is needed. Additionally, please clarify whether the sample showed a normal distribution across the variables.
- Line 226: The study includes significantly more female participants than male participants. Since gender was found to be a significant predictor in regression models, this should be addressed in the Discussion section. Currently, it is neither explained nor mentioned as a limitation.
- Table 1: The t-test results were significant, whereas ANOVA did not show differences. Could this be due to a higher number of groups in the ANOVA? Please consider providing evidence or explanation for this discrepancy.
- Table 2: It would be beneficial to include cut-off scores, particularly for burnout, to contextualize the findings.
- The results section lacks a detailed breakdown of burnout subscales. This is a crucial omission since different burnout dimensions may be influenced differently by the explanatory variables. Providing these results would be more informative than the correlation table, which offers limited new insights compared to the regression analysis. If space constraints exist, I suggest replacing the correlation table with a more detailed analysis of the burnout subscales.
- Table 4: The asterisk (*) notation is used differently from the conventional practice, and the significance levels are not explicitly marked with symbols. This should be corrected for clarity.
- Lines 288-289: The sentence "In our study, forming organisational relationships based on trust, a subfactor of positive nursing organisational culture, was the factor with the strongest effect on burnout." seems inconsistent with the regression results, where grit was the strongest predictor. If this is the case, I recommend beginning the discussion with the grit findings instead.
- Line 329: A new paragraph should be started here to improve readability.
- Gender and marital status were significant predictors of burnout but are not discussed in the paper. This is a crucial omission, as demographic factors can have important implications for burnout interventions. Please address this in the discussion.
Author Response
Responses to Reviewer 1’s Comments
My co-author and I wish to re-submit our revised manuscript entitled “The effects of grit, emergency nursing competency, and positive nursing organisational culture on burnout among nurses in emergency department” with changes that thoroughly address your comments.
We thank you for your thoughtful suggestions and insights. The manuscript has benefited from your thorough feedback. We look forward to working with you to move this manuscript closer to publication in Behavioral Sciences.
The manuscript has been rechecked and the necessary changes have been made in accordance with your suggestions. The responses to all comments have been prepared and attached herewith/given below.
We appreciate your consideration.
Comments 1: Line 11: I recommend explicitly stating that the study is a "descriptive cross-sectional study" in the abstract to improve clarity for readers.
Response 1: Thank you for your careful recommendation. We have accordingly added the information regarding the study design - page 1, abstract, line 11.
“Abstract: This descriptive cross-sectional study investigated the relationship between grit, emergency”
Comments 2: Line 71: The sentence "One meta-analysis demonstrated that grit is associated with conscientiousness [19]." appears disconnected from the surrounding context. It would be beneficial to clarify its relevance to the study’s objectives.
Response 2: Thank you for your logical comment. Based on your advice, we reviewed the reference [19] and removed this point from our revised manuscript.
Comments 3: Lines 78-83: This paragraph would be more appropriately placed at the end of the introduction. Additionally, I recommend elaborating on the research questions and hypotheses in this section to create a clearer logical flow. Currently, the transition to the study’s aims is too brief and general (lines 133-135). More precise research questions or hypotheses should be explicitly stated.
Response 3: Thank you for your clear suggestion. Based on your recommendation, we have placed lines 78-83 at the end of the introduction - pages 3-4, paragraph 4, lines 131-139.
“Being provided ample opportunities to make intentional efforts toward a goal allows one to develop grit, and this strategy could help alleviate burnout (Lee et al., 2023) in ED nurses. Although grit is a personal characteristic, it can be shaped through education and experience (Khedr et al., 2023). Further, as organisational culture is a factor that changes over time (Aghaei & Asadi, 2020), it is vital to conduct research to understand the relationship between burnout and positive nursing organisational culture in recent ED nursing organisations, which have been affected by various infectious diseases. Therefore, to alleviate or prevent burnout in ED nurses, it is crucial to investigate the relationship between grit, emergency nursing competency, and positive organizational culture.”
Further, according to your suggestion, we have revised the text explaining the purpose of this study and presented the research questions – page 4, paragraph 2, lines 140-148.
“1.1. Aim
The aim of this study was to investigate the influence of ED nurses’ grit, emergency nursing competency, and positive nursing organisational culture on their burnout experiences. The research questions were as follows: (1) What are the levels of burnout experienced by ED nurses and the related differences according to their general characteristics? (2) What are the levels of grit, emergency nursing competency, and positive nursing organizational culture? (3) Is there a relationship between grit, emergency nursing competency, positive nursing organizational culture, and burnout? (4) What are the factors influencing burnout in ED nurses?”
Comments 4: Lines 184-190: The burnout questionnaire includes three subscales, but their detailed results are not presented later in the paper. Furthermore, it would be useful to indicate whether there are commonly used cut-off scores for the Maslach Burnout Inventory and how the studied population compares to these thresholds.
Response 4: Thank you for your valuable comment. Based on your recommendation, we have presented the results of the subscales in Tables 2 and 3. We used the total sum of the MBI scale (Korean version) as the dependent variable in the regression analysis (see Table 4) - pages 7-8, Table 2-3, lines 257-259, 269-272.
Furthermore, according to Mind Garden, Inc., which holds the copyright to the MBI instrument, up until the third edition of the MBI published in 1996, a cut-off score was presented that classified the top third of the standard population as having high levels of emotional exhaustion, by arbitrarily dividing them into thirds. However, starting with the fourth edition of the MBI Manual published in 2016, the cut-off score has been removed from all MBI-related materials due to a lack of validity in diagnosing burnout.
[https://www.mindgarden.com/117-maslach-burnout-inventory-mbi#horizontalTab4]
[Maslach Burnout Inventory (MBI) - Assessments, Tests | Mind Garden - Mind Garden]
Further, as there were no suggested cut-off scores in the Korean version of the MBI, we did not present them. We ask for your kind understanding.
Comments 5: Lines 195-200: The methodology mentions that data collection occurred during nurses’ shift times and that participants received a small gift. These factors could have influenced the results. I suggest discussing this potential bias explicitly in the limitations section.
Response 5: Thank you for your insightful comment. Based on your suggestion, we have added a description about sampling bias in the limitations section - page 11, paragraph 2, lines 395-398.
“in later studies. Further, because we did not use random sampling to select the participants and provided a small gift as a token of appreciation for their engagement, we cannot exclude the possibility of sampling bias. Moreover, the nurse population in Korea generally consists of a large number of female nurses, which could again lead to sampling bias.”
Comments 6: Line 203: Please specify the exact type of ANOVA used. It is likely one-way ANOVA, but confirmation is needed. Additionally, please clarify whether the sample showed a normal distribution across the variables.
Response 6: Thank you for your comment. Based on your advice, we have mentioned the type of ANOVA, one-way ANOVA. Additionally, we have added details on the normal distribution test result, the Shapiro-Wilk test, because the sample size was under 2,000 - page 5, paragraph 5, lines 215-222.
“The collected data were analysed using descriptive statistics, independent t-tests, and one-way ANOVA, and the software used was SPSS Statistics 29.0. The correlations of burnout with grit, emergency nursing competency, and positive nursing organisational culture were analysed using Pearson’s correlation. To identify factors affecting burnout, we performed hierarchical multiple regression analysis to control for the general characteristics of participants from the seven study hospitals. The normal distribution of burnout based on the general characteristics of the participants was analysed using the Shapiro-Wilk test. The normal distribution was confirmed at p>0.05.”
Comments 7: Line 226: The study includes significantly more female participants than male participants. Since gender was found to be a significant predictor in regression models, this should be addressed in the Discussion section. Currently, it is neither explained nor mentioned as a limitation.
Response 7: Thank you for your valuable suggestion. Based on your comment, we have explained the aspects related to gender and marital status in the discussion section and noted the related limitations as well - page 10, paragraphs 4, lines 361-377.
“Although gender and marital status were significant factors in the univariate analysis in Model 1, they were not significant in the final multivariate analysis in Model 3 for burnout. This result is consistent with the findings of a previous systematic review focused on emergency department healthcare workers, which revealed that burnout experiences did not significantly differ based on gender (Alanazy & Alruwaili, 2023). However, different results have been reported in studies involving Korean clinical nurses (Cha & Baek, 2023), city hospital nurses in Turkey (Aslan Savcı & Bayraktar, 2025), and Chinese clinical nurses (Xie et al., 2021). Although the evidence on gender is inconsistent, this result is not entirely inexplicable; globally, female nurses are likely more vulnerable to and influenced by burnout, due to diverse reasons such as societal roles and expectations, emotional factors, and family responsibilities, among others. Therefore, female nurses should be considered as an important population in burnout management efforts. Additionally, the finding on marital status is consistent with the results of previous research; higher scores were reported in the unmarried participant group, but it was not a predictor of burnout in univariate analysis (Cha & Baek, 2023), and there were no clear differences associated with burnout (Lim & Shin, 2022; Xie et al., 2021). Further research exploring the link between marital status and burnout is recommended.”
- page 11, paragraphs 2, lines 395-398.
“in later studies. Further, because we did not use random sampling to select the participants and provided a small gift as a token of appreciation for their engagement, we cannot exclude the possibility of sampling bias. Moreover, the nurse population in Korea generally consists of a large number of female nurses, which could again lead to sampling bias.”
Comments 8: Table 1: The t-test results were significant, whereas ANOVA did not show differences. Could this be due to a higher number of groups in the ANOVA? Please consider providing evidence or explanation for this discrepancy.
Response 8: Thank you for your comment and question. We divided the groups based on previous research (Cha & Baek, 2023; Lim & Shin, 2022). The factors of gender, marital status, education level, and emergency medical centre type were organized by unique nominalization. The factors of age, clinical career, and emergency department career were categorized based on a previous study (Cha & Baek, 2022), which presented differences in burnout according to these variables. Monthly income was surveyed based on four levels: <2 million won, 2−3 million won, 3−4 million won, >4 million won. Because the size of the <2 million won monthly income group was very small (n=1, 0.006%), we included this group to the <3 million won group. Further, as the >4 million won monthly income group constituted 11.4% of the total, this group was not small enough to require nonparametric statistics, and thus it was maintained.
In addition, there was no difference in the results when the groups in ANOVA were divided similarly and analyzed with a t-test. So we decided to use the group classification used in ANOVA analysis.
Comments 9: Table 2: It would be beneficial to include cut-off scores, particularly for burnout, to contextualize the findings.
Response 9: Thank you for your valuable comment. Based on your advice, we have included the subscale scores. Please see our Response 4 above explaining why the cut-off scores were not included.
Comments 10: The results section lacks a detailed breakdown of burnout subscales. This is a crucial omission since different burnout dimensions may be influenced differently by the explanatory variables. Providing these results would be more informative than the correlation table, which offers limited new insights compared to the regression analysis. If space constraints exist, I suggest replacing the correlation table with a more detailed analysis of the burnout subscales.
Response 10: Thank you for your comment. Based on your advice, we have presented detailed results for the burnout subscales in Tables 2 and 3. We have also added the correlation results for the burnout subscales in Table 3 - page 7, paragraph 1, Table 2, lines 254-259.
- page 7, Table 3, lines 269-272.
Comments 11: Table 4: The asterisk (*) notation is used differently from the conventional practice, and the significance levels are not explicitly marked with symbols. This should be corrected for clarity.
Response 11: Thank you for your attention to detail. Based on your advice, we have changed the asterisk to (†) in Table 4 - pages 8-9, table 4, lines 296-300
Comments 12: Lines 288-289: The sentence "In our study, forming organisational relationships based on trust, a subfactor of positive nursing organisational culture, was the factor with the strongest effect on burnout." seems inconsistent with the regression results, where grit was the strongest predictor. If this is the case, I recommend beginning the discussion with the grit findings instead.
Response 12: Thank you for your suggestion. In regression analysis, the degree to which an independent variable affects a dependent variable is determined by the size of the standardized regression coefficient (β). Therefore, in this study, the standardized regression coefficient of relationships based on trust under PNOC is [β = -0.31] and the standardized regression coefficient of GRIT is [β = -0.29]; thus, we wish to maintain the current order of the discussion.
Comments 13: Line 329: A new paragraph should be started here to improve readability.
Response 13: Thank you for your thoughtful advice. Based on your suggestion, we have split the paragraph and started a new paragraph where indicated - page 10, paragraph 3, line 354.
“Emergency nursing competency, however, was not found to have a significant effect”
Comments 14: Gender and marital status were significant predictors of burnout but are not discussed in the paper. This is a crucial omission, as demographic factors can have important implications for burnout interventions. Please address this in the discussion.
Response 14: Thank you for your valuable suggestion. In this study, gender and marital status were significant in the univariate analysis in Model 1, but this was not the case in the multivariate regression analysis in Model 2; thus, finally, these general characteristics variables were not significant predictors in Model 3. We have added this explanation in the discussion based on each model step - page 10, paragraph 4, lines 361-377.
“Although gender and marital status were significant factors in the univariate analysis in Model 1, they were not significant in the final multivariate analysis in Model 3 for burnout. This result is consistent with the findings of a previous systematic review focused on emergency department healthcare workers, which revealed that burnout experiences did not significantly differ based on gender (Alanazy & Alruwaili, 2023). However, different results have been reported in studies involving Korean clinical nurses (Cha & Baek, 2023), city hospital nurses in Turkey (Aslan Savcı & Bayraktar, 2025), and Chinese clinical nurses (Xie et al., 2021). Although the evidence on gender is inconsistent, this result is not entirely inexplicable; globally, female nurses are likely more vulnerable to and influenced by burnout, due to diverse reasons such as societal roles and expectations, emotional factors, and family responsibilities, among others. Therefore, female nurses should be considered as an important population in burnout management efforts. Additionally, the finding on marital status is consistent with the results of previous research; higher scores were reported in the unmarried participant group, but it was not a predictor of burnout in univariate analysis (Cha & Baek, 2023), and there were no clear differences associated with burnout (Lim & Shin, 2022; Xie et al., 2021). Further research exploring the link between marital status and burnout is recommended.”
|
|
END OF RESPONSES TO COMMENTS
Please let me know if you have any other concerns or questions about it. Thank you.
Corresponding Author

Reviewer 2 Report
Comments and Suggestions for Authors
This paper is well-written, well-organized, and addresses an important topic in emergency nursing. The study provides valuable insights into the relationship between grit, emergency nursing competency, and organizational culture in mitigating burnout. While the manuscript is strong in its structure and analysis, I offer some suggestions to further refine certain sections and enhance clarity.
Introduction
- Lines 33–35: The statement “When overcrowding, while the quantity of care that needs to be provided to patients increases, if this overlaps with a period of short staffing, it can exacerbate the work burden due to increased responsibilities and longer shift times, resulting in burnout” suggests a direct cause-effect relationship. However, burnout is a complex phenomenon influenced by multiple factors. The authors should clarify this relationship and discuss potential mediators or moderating factors.
- Lines 80–81: The claim “Although grit is a personal characteristic, it can be changed through education and experience” is presented as a given, but personality traits like grit are not always easily modified. The authors should support this statement with empirical evidence or consider a more cautious phrasing.
- Overall Structure: The introduction is lengthy and contains mixed information. I recommend restructuring it to present key concepts concisely, prioritizing the most relevant factors. Creating a separate Aims and Hypotheses subsection would enhance clarity.
Methods – Data Analysis
- The authors should specify whether a normality test was conducted (e.g., Kolmogorov-Smirnov) to justify the choice of statistical tests.
Discussion & Further Considerations
-
Alternative Explanations: The discussion could benefit from a broader perspective on factors that mitigate burnout. While the study focuses on organizational culture and grit, other personal and psychological aspects likely contribute to reducing burnout. These include coping strategies, resilience, personality traits (e.g., optimism, conscientiousness), and spiritual beliefs.
- Individuals with higher resilience may have developed adaptive mechanisms that mitigate the impact of burnout, regardless of workplace interventions.
- Personality traits, such as optimism and conscientiousness, influence how individuals perceive and respond to stressors.
- Spiritual beliefs or mindfulness practices have been associated with lower burnout rates, particularly in emotionally demanding healthcare settings.
-
Contextualizing Findings: These observations align with prior research on palliative care and intensive care professionals, who regularly work with critically ill patients yet report varying levels of burnout. Studies conducted before the COVID-19 pandemic suggest that intrinsic psychological factors might explain well-being and job satisfaction independently of external work conditions. This indicates that the study’s findings should not be solely attributed to workplace factors.
-
Suggested References for Expansion: I recommend incorporating insights from studies that explore these dimensions:
- Dahò, M. (2021). An Exploration of the Emotive Experiences and the Representations of Female Care Providers Working in a Perinatal Hospice: A Pilot Qualitative Study. Clinical Neuropsychiatry.
- Harris, L. T. J. M. (2013). Caring and Coping: Exploring How Nurses Manage Workplace Stress. Journal of Hospice and Palliative Nursing, 15(8), 446–454.
- Bruce, A., & Davies, B. (2005). Mindfulness in Hospice Care: Practicing Meditation-in-Action. Qualitative Health Research, 15(10), 1329–1344.
- Sinclair, S. (2011). Impact of Death and Dying on the Personal Lives and Practices of Palliative and Hospice Care Professionals. Canadian Medical Association Journal, 183(2), 180–187.
Author Response
Responses to Reviewer 2’s Comments
My co-author and I wish to re-submit our revised manuscript entitled “The effects of grit, emergency nursing competency, and positive nursing organisational culture on burnout among nurses in emergency department” with changes that thoroughly address your comments.
We thank you for your thoughtful suggestions and insights. The manuscript has benefited from your thorough feedback. We look forward to working with you to move this manuscript closer to publication in Behavioral Sciences.
The manuscript has been rechecked and the necessary changes have been made in accordance with your suggestions. The responses to all comments have been prepared and attached herewith/given below.
We appreciate your consideration.
Comments 1: Lines 33–35: The statement “When overcrowding, while the quantity of care that needs to be provided to patients increases, if this overlaps with a period of short staffing, it can exacerbate the work burden due to increased responsibilities and longer shift times, resulting in burnout” suggests a direct cause-effect relationship. However, burnout is a complex phenomenon influenced by multiple factors. The authors should clarify this relationship and discuss potential mediators or moderating factors.
Response 1: Thank you for your careful review. We have revised this sentence about ED nurses’ burnout experiences. We have described this as a nursing problem of burnout among ED nurses, avoiding suggestions of a cause-effect relationship of burnout and the environmental situation in the ED. Further, in this study, we did not consider mediators or moderators of burnout, so we have not described these factors to avoid possible confusion regarding the purpose of the study - page 1, paragraph 1, lines 33-36.
“The emergency department (ED) is characterised by overcrowding due to an endless stream of patients (Munn et al., 2025). In this context of overcrowding, ED nurses often encounter challenges such as time constraints, increased responsibilities, and longer shift times; they may thus experience burnout and frustration at work (Hetherington et al., 2024).”
Comments 2: Lines 80–81: The claim “Although grit is a personal characteristic, it can be changed through education and experience” is presented as a given, but personality traits like grit are not always easily modified. The authors should support this statement with empirical evidence or consider a more cautious phrasing.
Response 2: Thank you for your logical suggestion. Based on your advice, we have revised the text, providing empirical evidence - page 3, paragraph 4, lines 131-134.
“Being provided ample opportunities to make intentional efforts toward a goal allows one to develop grit, and this strategy could help alleviate burnout (Lee et al., 2023) in ED nurses. Although grit is a personal characteristic, it can be shaped through education and experience (Khedr et al., 2023).”
Comments 3: Overall Structure: The introduction is lengthy and contains mixed information. I recommend restructuring it to present key concepts concisely, prioritizing the most relevant factors. Creating a separate Aims and Hypotheses subsection would enhance clarity.
Response 3: Thank you for your careful review. Based on your suggestion and that of Reviewer 1, we have reconstructed the introduction section, moving some information toward the end of this section. We have also added a subsection presenting the study aim and described the research questions - page 4, paragraph 2, lines 140-148.
“1.1. Aim
The aim of this study was to investigate the influence of ED nurses’ grit, emergency nursing competency, and positive nursing organisational culture on their burnout experiences. The research questions were as follows: (1) What are the levels of burnout experienced by ED nurses and the related differences according to their general characteristics? (2) What are the levels of grit, emergency nursing competency, and positive nursing organizational culture? (3) Is there a relationship between grit, emergency nursing competency, positive nursing organizational culture, and burnout? (4) What are the factors influencing burnout in ED nurses?”
Comments 4: The authors should specify whether a normality test was conducted (e.g., Kolmogorov-Smirnov) to justify the choice of statistical tests.
Response 4: Thank you for your pertinent comment. We have added information about the normal distribution test results, the Shapiro-Wilk test, because the sample size was under 2,000 - page 5, paragraph 5, lines 215-222.
“The collected data were analysed using descriptive statistics, independent t-tests, and one-way ANOVA, and the software used was SPSS Statistics 29.0. The correlations of burnout with grit, emergency nursing competency, and positive nursing organisational culture were analysed using Pearson’s correlation. To identify factors affecting burnout, we performed hierarchical multiple regression analysis to control for the general characteristics of participants from the seven study hospitals. The normal distribution of burnout based on the general characteristics of the participants was analysed using the Shapiro-Wilk test. The normal distribution was confirmed at p>0.05.”
Comments 5: Alternative Explanations - The discussion could benefit from a broader perspective on factors that mitigate burnout. While the study focuses on organizational culture and grit, other personal and psychological aspects likely contribute to reducing burnout. These include coping strategies, resilience, personality traits (e.g., optimism, conscientiousness), and spiritual beliefs. - Individuals with higher resilience may have developed adaptive mechanisms that mitigate the impact of burnout, regardless of workplace interventions.
Response 5: Thank you for your careful review. We have added details based on our study results considering the variables of gender and marital status under the general characteristics. Further, we have also mentioned the impact of resilience as advised - page 10, paragraph 1, lines 00-00.
Comments 6: Contextualizing Findings: These observations align with prior research on palliative care and intensive care professionals, who regularly work with critically ill patients yet report varying levels of burnout. Studies conducted before the COVID-19 pandemic suggest that intrinsic psychological factors might explain well-being and job satisfaction independently of external work conditions. This indicates that the study’s findings should not be solely attributed to workplace factors.
Response 6: Thank you for your suggestion. We have now mentioned the importance of intrinsic psychological factors in the discussion. We have described it only briefly, because we believe that this may obscure the results, since organizational culture was the second most important factor in our study. We hope that you will take this into consideration - page 10, paragraph 4, lines 361-377.
“Although gender and marital status were significant factors in the univariate analysis in Model 1, they were not significant in the final multivariate analysis in Model 3 for burnout. This result is consistent with the findings of a previous systematic review focused on emergency department healthcare workers, which revealed that burnout experiences did not significantly differ based on gender (Alanazy & Alruwaili, 2023). However, different results have been reported in studies involving Korean clinical nurses (Cha & Baek, 2023), city hospital nurses in Turkey (Aslan Savcı & Bayraktar, 2025), and Chinese clinical nurses (Xie et al., 2021). Although the evidence on gender is inconsistent, this result is not entirely inexplicable; globally, female nurses are likely more vulnerable to and influenced by burnout, due to diverse reasons such as societal roles and expectations, emotional factors, and family responsibilities, among others. Therefore, female nurses should be considered as an important population in burnout management efforts. Additionally, the finding on marital status is consistent with the results of previous research; higher scores were reported in the unmarried participant group, but it was not a predictor of burnout in univariate analysis (Cha & Baek, 2023), and there were no clear differences associated with burnout (Lim & Shin, 2022; Xie et al., 2021). Further research exploring the link between marital status and burnout is recommended.”
- page 10, paragraph 2, lines 350-353.
“Further, individuals with higher resilience might be able to develop adaptive mechanisms; thus, resilience-based intervention from nursing student would impacted to mitigate the burnout on nurses (Khedr et al., 2023).”
Comments 7: Suggested References for Expansion: I recommend incorporating insights from studies that explore these dimensions:
Dahò, M. (2021). An Exploration of the Emotive Experiences and the Representations of Female Care Providers Working in a Perinatal Hospice: A Pilot Qualitative Study. Clinical Neuropsychiatry.
Harris, L. T. J. M. (2013). Caring and Coping: Exploring How Nurses Manage Workplace Stress. Journal of Hospice and Palliative Nursing, 15(8), 446–454.
Bruce, A., & Davies, B. (2005). Mindfulness in Hospice Care: Practicing Meditation-in-Action. Qualitative Health Research, 15(10), 1329–1344.
Sinclair, S. (2011). Impact of Death and Dying on the Personal Lives and Practices of Palliative and Hospice Care Professionals. Canadian Medical Association Journal, 183(2), 180–187.
Response 7: Thank you for your kind suggestion. We have cited these references in the discussion section for supporting the point on the importance of intrinsic psychological factors- page 11, paragraph 1, lines 383-386.
“Additionally, we considered the intrinsic factors that might explain the varying levels of burnout reported in previous research (Bruce & Davies, 2005; Dahò, 2021; Harris, 2013; Sinclair, 2011). Accordingly, workplace factors should be considered along with intrinsic psychological factors.”
|
|
END OF RESPONSES TO COMMENTS
Please let me know if you have any other concerns or questions about it. Thank you.
Corresponding Author

Round 2
Reviewer 1 Report
Comments and Suggestions for Authors
Thank You for considering my suggestions, the manuscript has been improved much.
Author Response
Response to Academic Editor Notes
My co-author and I wish to re-submit our revised manuscript entitled “The effects of grit, emergency nursing competency, and positive nursing organizational culture on burnout among nurses in emergency department” We have incorporated changes to thoroughly address your comments.
We thank you for your insightful suggestions and insights. The manuscript has benefited from your thorough feedback. We look forward to working with you to move this manuscript closer to publication in Behavioral Sciences.
The manuscript has been rechecked and the necessary changes have been made in accordance with your suggestions. The responses to all comments have been prepared and attached herewith/given below.
We appreciate your consideration.
Note 1: Can you check the sub-headings found in STROBE checklist.
Response 1: Thank you for this important recommendation. We have checked the sub-headings found in the STROBE checklist and revised our manuscript’s sub-headings accordingly. To this end, we have revised the subheadings under Methods and also added ‘Limitations’ as a sub-heading. Our revisions are indicated below.
- page 4, paragraph 2, line 140.
1.1. Objectives
- page 4, paragraph 3, lines 149-150.
- Methods
2.1. Study design
- page 4, paragraph 5, line 163.
2.3. Variables and measurement
- page 11, paragraph 3, line 392.
4.1. Limitations
Note 2: Ensure the methods section of your article makes reference to any missing.
Response 1: Thank you for your valuable recommendation. We have added the missing reference for positive nursing organisational culture on page 5, paragraph 2, line 193.
“Each item is rated on a 5-point scale, with higher scores indicating a more positive perception of nursing organisational culture. In terms of reliability, at the time of development (Kim & Kim, 2021)”
